# Computed Tomographic Hepatic Volumetry in Dogs with Congenital Portosystemic Shunts

**DOI:** 10.3390/vetsci11090390

**Published:** 2024-08-24

**Authors:** Hitomi Kurihara, George Moore, Masahiro Murakami

**Affiliations:** 1Department of Veterinary Clinical Sciences, College of Veterinary Medicine, Purdue University, West Lafayette, IN 47907, USA; 2Department of Veterinary Administration, College of Veterinary Medicine, Purdue University, West Lafayette, IN 47907, USA

**Keywords:** liver volumetry, vascular anomaly, microhepatia

## Abstract

**Simple Summary:**

The purpose of this study was to evaluate liver size differences in dogs with various types of congenital portosystemic shunts (PSSs) using computed tomographic hepatic volumetry (CTHV). PSSs can cause the liver to be small (microhepatia) and less functional due to blood bypassing the liver via the shunting vessel. The study measured liver volumes in dogs with different types of PSSs (intrahepatic (IH), extrahepatic portocaval (EHPC), extrahepatic portoazygos (EHPA), and extrahepatic portophrenic (EHPP)) in dogs without prior surgery. The study found that while the normalized liver volume (nLV) was similar across PSS types, the severity of microhepatia was more pronounced in dogs with EHPC and EHPA shunts compared with those with IH shunts. In addition, smaller dogs had more severe microhepatia, and liver volume decreased with age. These findings suggest that the severity of microhepatia varies with shunt type and body size and highlight the need for further research to truly understand the presence of microhepatia in dogs with PSSs. This study adds to the understanding of liver volume changes in dogs with PSSs and may help veterinarians provide better care for affected animals.

**Abstract:**

CTHV is a non-invasive and accurate method for assessing liver volume in dogs. CTHV has not been studied in each type of extrahepatic PSS in dogs. This study aimed to use CTHV to compare liver volumes in dogs with different types of PSSs that had been confirmed by computed tomography angiography. Dogs with PSSs were retrospectively included and categorized into IH, EHPC, EHPA, or EHPP shunt groups. Manual CTHV was performed, and the normalized liver volume (nLV) and the difference in nLV from the estimated liver volume calculated based on body weight (LV%diff) was calculated. The study included 57 dogs: 20 IH, 21 EHPC, 9 EHPA, and 7 EHPP. The median nLV (cm^3^/kg) and LV%diff (%) for each group were as follows: IH 17.3 (−40.4%); EHPC 16.9 (−60.3%); EHPA 15.1 (−56.7%); and EHPP 17.2 (−59.2%), respectively. There were no significant differences in nLV among the PSS types. However, LV%diff was significantly more pronounced in the EHPC and EHPA groups compared with the IH group. Additionally, smaller dogs exhibited more severe microhepatia, with a significant positive correlation between LV%diff and body weight (*p* < 0.01). These findings suggest that microhepatia severity varies by shunt type and is more severe in smaller dogs, highlighting the need for further research to understand the underlying mechanisms.

## 1. Introduction

Computed tomographic hepatic volumetry (CTHV) is a non-invasive volumetry method used in humans for the pre-operative evaluation of liver size with highly accurate volume predictions [1]. In recent years, CTHV has also been used in veterinary medicine, including for dogs with extrahepatic portosystemic shunts (PSSs), both prior to and following shunt correction [2,3,4]. Several reports have demonstrated that CTHV can identify pre-operative microhepatia and post-operative increases in liver size in dogs with PSSs [2,3,4].

Congenital PSSs in dogs can result in various clinical signs and laboratory abnormalities. This is due to the portal blood bypassing the liver and entering the systemic circulation via a shunt vessel, which leads to a small liver volume and impaired metabolism [4,5,6,7,8]. The presence of microhepatia, renomegaly, and uroliths has been shown to have a high positive predictive value (96%) for the presence of congenital PSSs in dogs [9]. Our recent studies have reported that kidney size varies among dogs with different types of PSSs, with severe renomegaly being more commonly found in dogs with intrahepatic (IH) and extrahepatic portocaval (EHPC) shunts, less common in dogs with extrahepatic portophrenic (EHPP) shunts, and uncommon in dogs with extrahepatic portoazygos (EHPA) shunts [10]. It has been proposed that the volume of blood shunted and the severity of liver dysfunction may influence the development of renomegaly in dogs with PSSs [10,11,12,13,14]. Therefore, it is suspected that the liver volume or degree of microhepatia may also differ among different types of PSSs, with the liver volume being smaller in IH and EHPC and larger in EHPA and EHPP shunt groups.

In recent research, we established a less time-consuming manual CTHV method, which exhibits high accuracy and low interobserver variability when employing ≥ 20 slices [15]. Body weight has been reported to be a preferred internal control for normalizing CT-derived liver volumes when using this method in dogs without liver disease [16]. Although a study has examined the pre- and postoperative changes in liver volume using CTHV in dogs with various types of PSSs [4], a comparison of liver volumes among different types of PSSs has yet to be investigated. In the previous study, the age at diagnosis differed among the different types of PSSs, and an association between kidney size and age was suggested [10]. However, no study to date has described the association between liver volume and age in dogs with PSSs.

Therefore, the primary objective of this study was to measure liver volumes using CTHV in dogs with different congenital PSS types (IH, EHPC, EHPA, and EHPP) before surgical correction, calculate the normalized liver volume (nLV), assess the degree of microhepatia using estimated liver volumes, and compare these parameters among different PSS types. Based on previous findings on kidney size [10], we hypothesized that the liver size would be the smallest and the microhepatia most severe in dogs with EHPC and IH shunts, the largest with mild microhepatia in dogs with EHPA shunts, and intermediate for both liver size and microhepatia in dogs with EHPP shunts. Additionally, the secondary objective was to analyze correlations between liver volumes or the degree of microhepatia and age or body weight in dogs with PSSs. We hypothesized that age would positively correlate with the nLV and degree of microhepatia, while body weight would not show such correlations.

## 2. Materials and Methods

### 2.1. Experimental Design and Case Selection Criteria

This retrospective case series study was performed using data from the Purdue University Veterinary Teaching Hospital (PUVTH) medical records database. Medical records were searched for dogs that underwent abdominal computed tomography (CT) between 1 January 2016 and 31 December 2020 and that were diagnosed with congenital portosystemic shunts (PSSs) by CT angiography.

CT images were independently reviewed by an American College of Veterinary Radiology (ACVR) board-certified radiologist (M.M.) who was blinded to the clinical information or diagnosis of the cases, and the type of PSS was classified. Categories included intrahepatic (IH), extrahepatic portocaval (EHPC), extrahepatic portoazygos (EHPA), and extrahepatic portophrenic (EHPP) shunts. Extrahepatic portosystemic shunts with shunt vessels connecting the portal vein and the caudal vena cava were classified as EHPC shunts, except when the shunt vessel joined the caudal vena cava cranial to the liver, which was considered an EHPP shunt. Extrahepatic shunts with vessels terminating in the azygos vein were classified as EHPA shunts. Dogs with any hepatic disease unrelated to PSSs that could affect the liver volumetry, including hepatic malignant neoplasia, were considered for exclusion. However, none of the dogs exhibited such abnormalities; thus, no exclusions were made. Only pre-contrast studies were included for further CT hepatic volumetry.

Patient information such as breed, age, and body weight were also recorded for each case.

### 2.2. CT Acquisition Technical Parameters

All CT studies were performed using a 64-slice multidetector CT machine (Light Speed VCT, GE Medical Systems Inc., Waukesha, WI, USA) with images acquired in the transverse plane using the following image acquisition parameters: helical scan mode, 100–120 kVp, 240–340 mA, slice thickness = 0.625–2.5 mm, tube rotation time = 1 s, pitch = 1, matrix = 512 × 512, and detail algorithm. Dogs were positioned in sternal recumbency under sedation or general anesthesia.

### 2.3. Computed Tomographic Hepatic Volumetry (CTHV) Analysis

CTHV was performed by a veterinarian (H.K.) who was being trained to perform CTHV under the supervision of a board-certified radiologist (M.M.). The analysis was performed using a DICOM viewer (Horos 64-bit, version 3.3.6) according to a previously published method [15]. For all dogs, the window width was set to 350 HU and the window level to 40 HU. Liver segmentation was performed by manually drawing the operator-defined region of interest (ROI) on pre-contrast transverse images of the entire liver. This process extended from the cranial border of the liver at the diaphragm to the most caudal borders of the liver adjacent to the right kidney and spleen.

The ROIs included the hepatic vessels within the liver parenchyma but excluded the gallbladder, visible hepatic lobe fissures, and vessels outside the liver parenchymal margin (Figure 1). After manually drawing the ROIs on more than 20 slices of the liver parenchyma, the CT-derived liver volume was calculated using the following formula to estimate liver volume: Σ {each slice area (cm^2^) × slice thickness (cm)} × total number of slices of hepatic parenchyma/number of slices [15]. The CT-derived liver volume was then normalized to the body weight (kg) of each dog to calculate the normalized CT-derived liver volume (nLV).

### 2.4. Statistical Analysis

The Shapiro–Wilk normality test was performed on the results of the nLV in each subgroup, and all the subgroup results indicated a non-parametric distribution of the nLVs. Consequently, the median and range were reported for the nLVs and the LV%diff.

The median nLV (cm^3^/kg; range) was calculated for each of the four groups: IH, EHPC, EHPA, and EHPP. The estimated liver volume for each dog was calculated using a previously reported model that demonstrates the relationship between the CT-derived liver volume and body weight in dogs without liver disease. This model is given as (CT-derived liver volume = 19 × (body weight; kg) + 97) [16]. The percentage difference between the CT-derived liver volume and the estimated liver volume (LV%diff) was then calculated for each dog. The median and range of the LV%diff was then recorded for each PSS group.

Since the data were non-parametric, the Kruskal–Wallis test was used to evaluate significant differences in nLVs and LV%diffs between the different types of PSSs. In cases where a significant difference was observed in the Kruskal–Wallis test, the post hoc Steel–Dwass test with Bonferroni correction was used to compare the median nLVs of the four groups. The significance level was set at *p* < 0.05.

Correlations between nLVs and age, LV%diffs and age, nLVs and body weight, and LV%diffs and body weight were evaluated using the Spearman’s rank correlation coefficient. In addition, to visually understand the distribution of LV%diffs by age and body weight, scatter plots were generated for all the PSS dogs.

## 3. Results

### 3.1. Animal Profiles

Fifty-seven dogs were included in the study, with 20 in the IH group, 21 in the EHPC group, 9 in the EHPA group, and 7 in the EHPP group. The breeds included 10 mixed breed dogs, 9 Golden Retrievers, 8 Yorkshire Terriers, 6 Labrador Retrievers, 4 Pugs, 2 Maltese, 2 Miniature Schnauzers, and 1 each of American Cocker Spaniel, American Eskimo, Beagle, Bernese Mountain Dog, Border Collie, Cockapoo, Goldendoodle, Irish Setter, Irish Wolfhound, Labradoodle, Lhasa Apso, Miniature Pinscher, Norwegian Elkhound, Pembroke Welsh Corgi, Toy Fox Terrier, and Welsh Terrier. The median body weights were 7.7 kg (range 0.8 to 55.1 kg) for all the PSS dogs and 13.1 kg (range 4.7 to 55.1 kg) for the IH, 5.0 kg (range 0.8 to 24.8 kg) for the EHPC, 6.5 kg (range 1.4 to 20.4 kg) for the EHPA, and 5.4 kg (range 2.4 to 24.7 kg) for the EHPP groups. The median age was 8 months (range 1 to 89 months) for all the PSS dogs, 5 months for the IH group, 8 months for the EHPC group, 33 months for the EHPA group, and 26 months for the EHPP group.

### 3.2. Computed Tomographic Hepatic Volumetry (CTHV)

The median (range) nLV of all PSS dogs was 16.8 (8.1 to 37.6) cm^3^/kg. The median nLV for each group was as follows: IH = 17.3 (8.1 to 21.6) cm^3^/kg; EHPC = 16.9 (9.3 to 28.0) cm^3^/kg; EHPA = 15.1 (10.6 to 37.6) cm^3^/kg; and EHPP = 17.2 (10.8 to 27.4) cm^3^/kg (Figure 2). The median (range) LV%diff for all the PSS dogs and for each group was as follows: all PSS dogs = −55.5 (−81.9 to −15.1)%; IH = −40.4 (−65.6 to −15.1)%; EHPC = −60.3 (−81.9 to −30.3)%; EHPA = −56.7 (−63.0 to −43.7)%; and EHPP = −59.2 (−71.1 to −25.8)% (Figure 3). No significant differences in the nLVs among the different shunt types were observed using the Kruskal–Wallis test (*p* = 0.49). However, significant differences were present in the LV%diffs (*p* < 0.01). The post hoc Steel–Dwass test revealed significant differences between the EHPC and IH groups (*p* < 0.01) and between the EHPA and IH groups (*p* = 0.03).

Spearman’s correlation analysis revealed a significant negative correlation between nLVs and age (*p* < 0.01, rho = −0.43; Figure 4A). However, there was no significant correlation between LV%diffs and age (*p* = 0.27; Figure 4B). A significant negative correlation was also found between nLVs and body weight (*p* < 0.01, rho = −0.56; Figure 4C), suggesting that larger body weights are associated with smaller liver volumes. Additionally, a significant positive correlation was identified between LV%diffs and body weight (*p* < 0.01, rho = 0.65; Figure 4D), indicating that the percentage difference in liver volume increases with body weight.

In the entire group of dogs with PSSs, 24 dogs (24/57, 42.1%) had an nLV smaller than previously reported mean nLV in dogs without liver disease [16]. This included 7/20 (35.0%) in the IH group, 8/21 (38.1%) in the EHPC group, 6/9 (66.7%) in the EHPA group, and 3/7 (42.9%) in the EHPP group. Notably, dogs aged 26 months or younger exhibited a wide range of nLVs, from very small to large, while dogs older than 26 months consistently had small nLVs, all below the previously reported mean nLV in dogs without liver disease (Figure 4A). Similarly, dogs aged 15 months or younger showed a wide range of LV%diffs from −81.9% to −15.1%, while dogs older than 15 months consistently had large negative LV%diffs, which were all greater than a negative 40% difference (Figure 4B).

## 4. Discussion

The present study aimed to investigate the differences in nLV and LV%diff among dogs with various types of congenital PSSs. Contrary to our initial hypothesis, we found no significant differences in nLV between shunt types. However, LV%diff was significantly lower in dogs with EHPC and EHPA shunts compared with those with IH shunts. Our second hypothesis was also partially refuted as we observed a mild negative correlation between nLV and age, no significant correlation between LV%diff and age, and moderate to strong negative correlations between nLV or LV%diff and body weight. These findings suggest that microhepatia is more severe in smaller dogs with extrahepatic portosystemic shunts, except in those with EHPP shunts, and that age does not significantly influence liver volume in these cases.

Assessing liver size through various imaging modalities, such as radiography, ultrasonography, and CT, is essential for understanding various hepatic diseases in dogs [16,17,18,19,20]. Radiography is the most commonly used clinical method for assessing liver size in dogs. In dogs, microhepatia is usually diagnosed by observing cranial displacement of the gastric axis on the radiograph. Although chronic inflammatory disease, fibrosis, cirrhosis, atrophy, and PSSs are commonly considered to be causes of microhepatia in dogs [17,18,19], the poor accuracy of radiographic evaluation of liver size, due to variations in body conformation, body condition, and the influence of respiratory timing, has resulted in an incomplete understanding of the presence of microhepatia in dogs [21,22]. CTHV is a more accurate method for measuring liver volume in dogs [23]. Previous research identified body weight as the optimal internal control, providing a tentative range for normalized liver volume (nLV) in healthy dogs, with a raw nLV of 22.1 cm^3^/kg (95% CI: 12.9–31.3 cm^3^/kg) and a regression model of volume = 19 × Body weight (kg) + 97 [16].

Microhepatia is a frequent finding in dogs with congenital PSSs, and reduced portal blood flow has been suggested as the primary cause. The bypassing of portal blood directly into the systemic circulation via shunting vessels reduces the delivery of hepatotropic factors, such as insulin and glucagon, to the liver parenchyma. These factors, which are derived from the gastrointestinal tract and pancreas, play an essential role in maintaining the liver parenchymal cell volume and function. As a result, reduced delivery of hepatotropic factors is likely to contribute to liver atrophy and parenchymal degenerative changes [24,25]. The extent of blood shunting or the shunt fraction may thus be pivotal in the development of microhepatia.

In the present study, dogs with PSSs exhibited generally smaller liver volumes, with the median nLV being lower than the previously reported mean nLV in dogs without liver disease, though still within the 95% CI [16]. It is noteworthy that no established definition specifies how small a liver must be to qualify as microhepatia. Nonetheless, some dogs in our study did not present with microhepatia. Specifically, seven dogs (four in the EHPC group, two in the EHPP group, and one in the EHPA group) had nLVs exceeding the mean nLV in dogs without liver disease. Moreover, one dog in the EHPA group had an nLV surpassing the upper limit of the 95% CI in dogs without liver disease [16]. Therefore, it is important to recognize that the absence of microhepatia does not necessarily rule out the presence of PSSs.

Previous research has not investigated the degree of microhepatia across different types of portosystemic shunts (PSSs), although the occurrence and severity of renomegaly, another complication of PSSs, have been reported to vary among PSS types in dogs [10]. In the present study, while there was no significant difference in the nLV among dogs with different PSS types, the degree of microhepatia, indicated by LV%diff, was significantly more severe in dogs with EHPC and EHPA shunts compared with those with IH shunts. This finding contradicts theoretical expectations, as IH shunts typically result in a higher shunt fraction than extrahepatic PSSs [26,27], leading to the anticipation of more severe microhepatia in IH cases.

It has been suggested that the shunt vessels in EHPA and EHPP shunts are partially compressed by the diaphragm during normal respiration and by gastric distension after eating, which could improve hepatic perfusion via the portal vein. Additionally, the azygos vein involved in EHPA is narrower and has less capacity than the caudal vena cava, potentially creating resistance to blood flow and inhibiting shunting from the portal system [28,29]. These factors are considered to contribute to the lower shunt fraction in dogs with EHPA and EHPP, which should, theoretically, result in less severe microhepatia in these dogs.

However, in the present study, more severe microhepatia was observed in the EHPA group compared with the IH group. Given the lack of comprehensive studies comparing shunt fractions among different PSS types in dogs and the absence of shunt fraction measurements via nuclear scintigraphy [29,30] or dynamic CT [27] in this study, it remains unclear if the observed differences in microhepatia severity are attributable to varying shunt fractions. Without a robust pathophysiological explanation, the significant differences in microhepatia severity among the different PSS types observed in this study warrant further investigation.

Our results demonstrated a negative correlation between nLV and age, indicating that older dogs tend to have smaller livers. Additionally, dogs in the EHPA group were the oldest among the PSS groups. This age factor could contribute to the more severe microhepatia observed in EHPA dogs. The smaller liver size in older dogs may be attributed to the chronicity of liver damage, which exacerbates microhepatia over time [5]. However, this hypothesis remains speculative due to the lack of histopathological examinations in our study. Future research should measure actual shunt fractions and include histopathological evaluations to clarify the contributions of chronic liver damage and shunt fraction to the development of microhepatia.

The degree of microhepatia also correlated with body weight, being significantly more severe in smaller dogs. This observation may be due to the prevalence of extrahepatic PSSs, which are more common in small dog breeds, while IH PSSs are more common in larger breeds [31]. In our study, the median body weight for dogs with IH shunts was 13.1 kg compared with 5.0 to 6.5 kg for those with extrahepatic PSSs. Since the shunt fraction does not fully explain these changes, it is possible that smaller dogs are more prone to developing severe microhepatia or that larger dogs exhibit milder microhepatia.

Another possible explanation for the more severe microhepatia in smaller dogs may be related to the method used in this study. The estimated liver volume was calculated using a formula from a previous study involving dogs weighing more than 6.5 kg [16]. This formula may overestimate the liver volume in dogs weighing less than 6.5 kg, resulting in a greater degree of negative LV%diffs in the small dogs in the current study. Future research should consider developing or validating a formula specifically for smaller dogs to ensure accurate liver volume estimations.

One of the limitations of this study is the lack of a standardized anesthesia or sedation protocol for all cases. Because this is a retrospective study, the included dogs received different anesthesia and sedation regimens based on their clinical need at the time of their respective CT studies. Variations in anesthesia protocols can affect physiological parameters such as hepatic blood flow and portal venous pressure, potentially affecting liver size measurements. The lack of a standardized protocol introduces an additional source of variability that may have affected the CTHV results. Future prospective studies using standardized anesthesia or sedation protocols would help minimize these confounding factors and provide more consistent data regarding hepatic volumetry in dogs with congenital portosystemic shunts.

## 5. Conclusions

In conclusion, this study demonstrates that while liver volumes do not differ significantly between PSS types, the severity of microhepatia, as indicated by the LV%diff, is more pronounced in dogs with EHPC and EHPA shunts compared with those with IH shunts. In addition, the normalized liver volume decreases with age and body weight; however, smaller dogs showed more severe microhepatia. These findings highlight the need for further research to understand the underlying mechanisms and to develop better diagnostic and therapeutic strategies for dogs with congenital PSSs.

## Figures and Tables

**Figure 1 vetsci-11-00390-f001:**
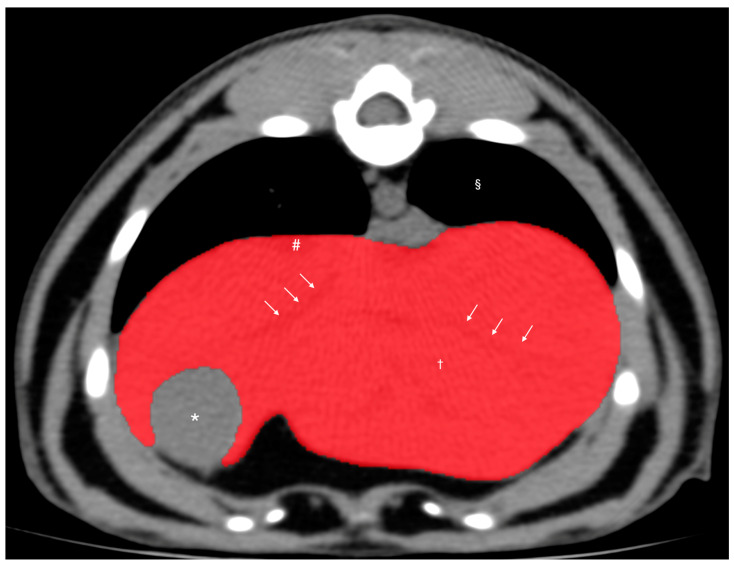
Pre-contrast transverse abdominal CT images used for CT hepatic volumetry in dogs. The segments of the liver were manually selected as the regions of interest (ROIs: highlighted in red). Note that the hepatic vessels within the liver parenchyma were included (white arrows). The gallbladder, hepatic lobe fissure, and hepatic vessels present outside the hepatic parenchyma were excluded. Window width, 350 HU; window level, 40 HU. Liver parenchyma (†), gallbladder (*), caudal vena cava (#), and pulmonary parenchyma (§).

**Figure 2 vetsci-11-00390-f002:**
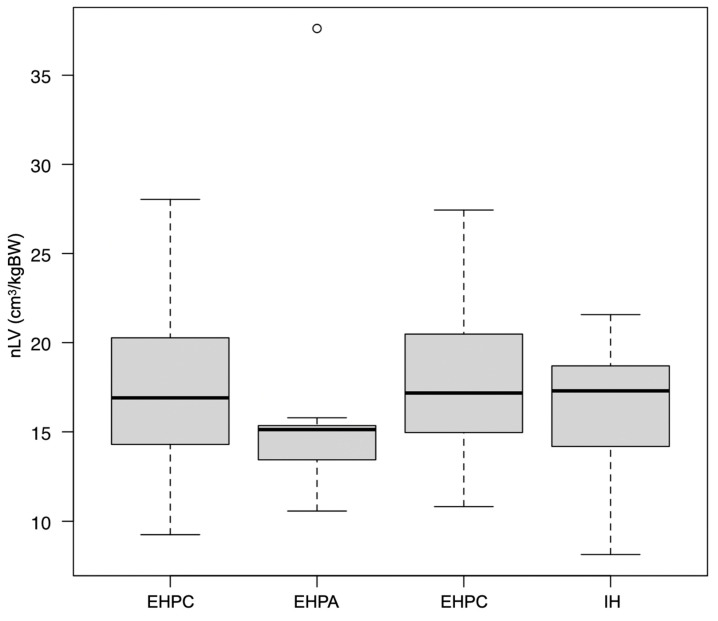
Box-and-whiskers plot of the normalized liver volume (nLV) in dogs with different shunt types. There were no significant differences in nLVs among the shunt types using the Kruskal–Wallis test. Each box represents the interquartile range. The horizontal bar through the box represents the median. Whiskers represent the range. The outlying data point is represented by a circle.

**Figure 3 vetsci-11-00390-f003:**
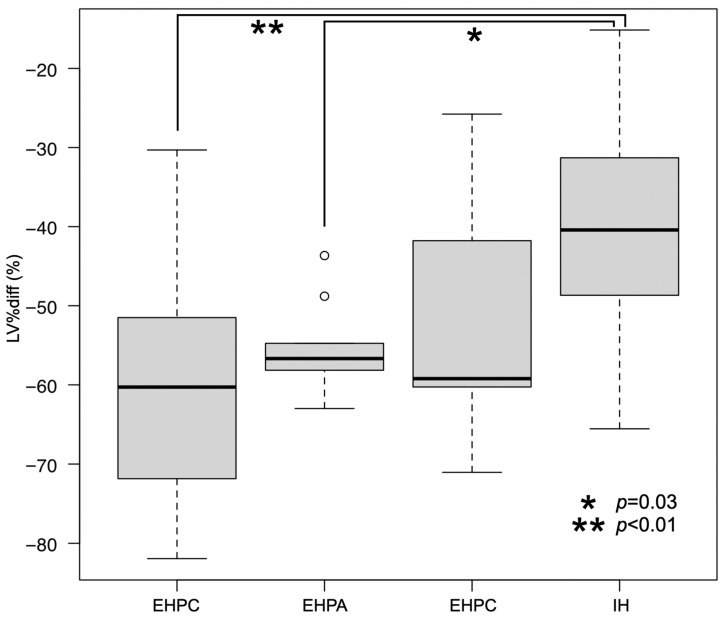
Median LV%diff with interquartile and 95% range in the dogs for each shunt type. There were significant differences in nLVs between the EHPC and IH (*p* < 0.01) and EHPA and IH (*p* = 0.03) groups using the Kruskal–Wallis test. Each box represents the interquartile range. The horizontal bar through the box represents the median. Whiskers represent the range. Outlying data points are represented by circles.

**Figure 4 vetsci-11-00390-f004:**
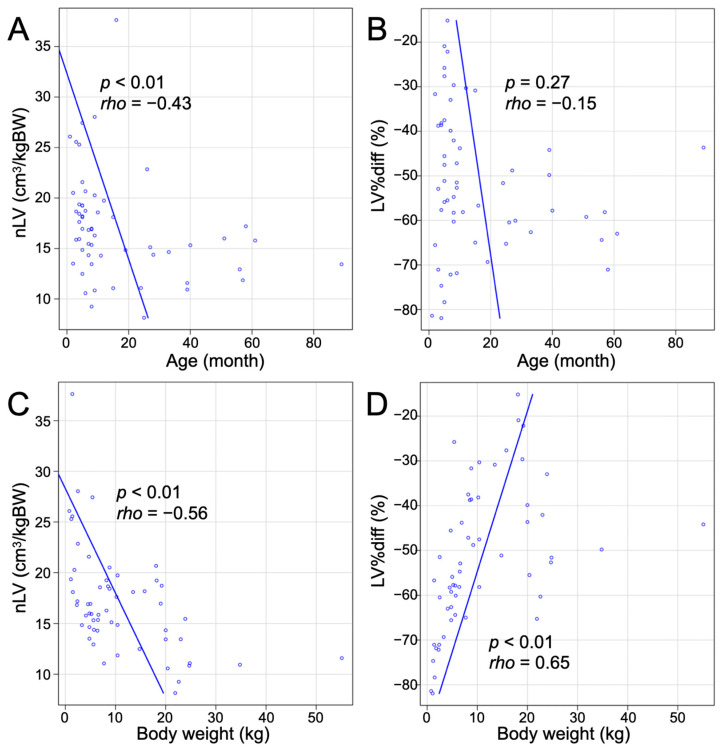
Correlation between nLV and LV%diff with age and body weight evaluated by Spearman’s correlation analysis. (**A**) Scatter plot showing the relationship between nLV and age. There is a significant negative correlation (*p* < 0.01, rho = −0.43). (**B**) Scatter plot showing the relationship between LV%diff and age. No significant correlation was found (*p* = 0.27). (**C**) Scatter plot showing the relationship between nLV and body weight. There is a significant negative correlation (*p* < 0.01, rho = −0.56). (**D**) Scatter plot showing the relationship between LV%diff and body weight. There is a significant positive correlation (*p* < 0.01, rho = 0.65).

## Data Availability

The original contributions presented in the study are included in the article; further inquiries can be directed to the corresponding author.

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
