# Peer review of "Computed Tomographic Hepatic Volumetry in Dogs with Congenital Portosystemic Shunts"

_vetsci, 2024, doi:10.3390/vetsci11090390_

Round 1
Reviewer 1 Report
Comments and Suggestions for Authors
I carefully read the manuscript titled ‘Computed tomographic hepatic volumetry in dogs with congenital portosystemic shunts.’ This study assessed liver size variations in dogs with different types of congenital portosystemic shunts through computed tomographic hepatic volumetry. Additionally, it explored the relationship between liver size and factors like age and body weight. The results are interesting and comprehensibly worded. The discussion is understandable and well-structured. A few comments and remarks follow below.
Quoted comments are provided below.
Line 25-26: Please indicate the examination with which the diagnosis and categorization of congenital portosystemic shunts was made. It is not clear that a CT scan was performed.
Line 32-33: Please indicate the age and body weight of dogs, an average value.
Line 40: Please refresh, non-invasive
Line 105-106: why wasn't the same anesthesia or sedation protocol followed for all animals? do you think that the use of different anesthetic drugs could affect the blood supply to the liver and therefore the total volume of the liver at the time of the examination?
Line 220-226: Except from x-rays, it is necessary to mention the ultrasound examination for the liver size evaluation and the Doppler examination for the imaging of vessels and assessment of blood flow volume.
Line 231-234: please add reference.I carefully read the manuscript titled ‘Computed tomographic hepatic volumetry in dogs with congenital portosystemic shunts.’ This study assessed liver size variations in dogs with different types of congenital portosystemic shunts through computed tomographic hepatic volumetry. Additionally, it explored the relationship between liver size and factors like age and body weight. The results are interesting and comprehensibly worded. The discussion is understandable and well-structured. A few comments and remarks follow below.
Quoted comments are provided below.
Line 25-26: Please indicate the examination with which the diagnosis and categorization of congenital portosystemic shunts was made. It is not clear that a CT scan was performed.
Line 32-33: Please indicate the age and body weight of dogs, an average value.
Line 40: Please refresh, non-invasive
Line 105-106: why wasn't the same anesthesia or sedation protocol followed for all animals? do you think that the use of different anesthetic drugs could affect the blood supply to the liver and therefore the total volume of the liver at the time of the examination?
Line 220-226: Except from x-rays, it is necessary to mention the ultrasound examination for the liver size evaluation and the Doppler examination for the imaging of vessels and assessment of blood flow volume.
Line 231-234: please add reference.I carefully read the manuscript titled ‘Computed tomographic hepatic volumetry in dogs with congenital portosystemic shunts.’ This study assessed liver size variations in dogs with different types of congenital portosystemic shunts through computed tomographic hepatic volumetry. Additionally, it explored the relationship between liver size and factors like age and body weight. The results are interesting and comprehensibly worded. The discussion is understandable and well-structured. A few comments and remarks follow below.
Quoted comments are provided below.
Line 25-26: Please indicate the examination with which the diagnosis and categorization of congenital portosystemic shunts was made. It is not clear that a CT scan was performed.
Line 32-33: Please indicate the age and body weight of dogs, an average value.
Line 40: Please refresh, non-invasive
Line 105-106: why wasn't the same anesthesia or sedation protocol followed for all animals? do you think that the use of different anesthetic drugs could affect the blood supply to the liver and therefore the total volume of the liver at the time of the examination?
Line 220-226: Except from x-rays, it is necessary to mention the ultrasound examination for the liver size evaluation and the Doppler examination for the imaging of vessels and assessment of blood flow volume.
Line 231-234: please add reference.I carefully read the manuscript titled ‘Computed tomographic hepatic volumetry in dogs with congenital portosystemic shunts.’ This study assessed liver size variations in dogs with different types of congenital portosystemic shunts through computed tomographic hepatic volumetry. Additionally, it explored the relationship between liver size and factors like age and body weight. The results are interesting and comprehensibly worded. The discussion is understandable and well-structured. A few comments and remarks follow below.
Quoted comments are provided below.
Line 25-26: Please indicate the examination with which the diagnosis and categorization of congenital portosystemic shunts was made. It is not clear that a CT scan was performed.
Line 32-33: Please indicate the age and body weight of dogs, an average value.
Line 40: Please refresh, non-invasive
Line 105-106: why wasn't the same anesthesia or sedation protocol followed for all animals? do you think that the use of different anesthetic drugs could affect the blood supply to the liver and therefore the total volume of the liver at the time of the examination?
Line 220-226: Except from x-rays, it is necessary to mention the ultrasound examination for the liver size evaluation and the Doppler examination for the imaging of vessels and assessment of blood flow volume.
Line 231-234: please add reference.I carefully read the manuscript titled ‘Computed tomographic hepatic volumetry in dogs with congenital portosystemic shunts.’ This study assessed liver size variations in dogs with different types of congenital portosystemic shunts through computed tomographic hepatic volumetry. Additionally, it explored the relationship between liver size and factors like age and body weight. The results are interesting and comprehensibly worded. The discussion is understandable and well-structured. A few comments and remarks follow below.
Quoted comments are provided below.
Line 25-26: Please indicate the examination with which the diagnosis and categorization of congenital portosystemic shunts was made. It is not clear that a CT scan was performed.
Line 32-33: Please indicate the age and body weight of dogs, an average value.
Line 40: Please refresh, non-invasive
Line 105-106: why wasn't the same anesthesia or sedation protocol followed for all animals? do you think that the use of different anesthetic drugs could affect the blood supply to the liver and therefore the total volume of the liver at the time of the examination?
Line 220-226: Except from x-rays, it is necessary to mention the ultrasound examination for the liver size evaluation and the Doppler examination for the imaging of vessels and assessment of blood flow volume.
Line 231-234: please add reference.I carefully read the manuscript titled ‘Computed tomographic hepatic volumetry in dogs with congenital portosystemic shunts.’ This study assessed liver size variations in dogs with different types of congenital portosystemic shunts through computed tomographic hepatic volumetry. Additionally, it explored the relationship between liver size and factors like age and body weight. The results are interesting and comprehensibly worded. The discussion is understandable and well-structured. A few comments and remarks follow below.
Quoted comments are provided below.
Line 25-26: Please indicate the examination with which the diagnosis and categorization of congenital portosystemic shunts was made. It is not clear that a CT scan was performed.
Line 32-33: Please indicate the age and body weight of dogs, an average value.
Line 40: Please refresh, non-invasive
Line 105-106: why wasn't the same anesthesia or sedation protocol followed for all animals? do you think that the use of different anesthetic drugs could affect the blood supply to the liver and therefore the total volume of the liver at the time of the examination?
Line 220-226: Except from x-rays, it is necessary to mention the ultrasound examination for the liver size evaluation and the Doppler examination for the imaging of vessels and assessment of blood flow volume.
Line 231-234: please add reference.I carefully read the manuscript titled ‘Computed tomographic hepatic volumetry in dogs with congenital portosystemic shunts.’ This study assessed liver size variations in dogs with different types of congenital portosystemic shunts through computed tomographic hepatic volumetry. Additionally, it explored the relationship between liver size and factors like age and body weight. The results are interesting and comprehensibly worded. The discussion is understandable and well-structured. A few comments and remarks follow below.
Quoted comments are provided below.
Line 25-26: Please indicate the examination with which the diagnosis and categorization of congenital portosystemic shunts was made. It is not clear that a CT scan was performed.
Line 32-33: Please indicate the age and body weight of dogs, an average value.
Line 40: Please refresh, non-invasive
Line 105-106: why wasn't the same anesthesia or sedation protocol followed for all animals? do you think that the use of different anesthetic drugs could affect the blood supply to the liver and therefore the total volume of the liver at the time of the examination?
Line 220-226: Except from x-rays, it is necessary to mention the ultrasound examination for the liver size evaluation and the Doppler examination for the imaging of vessels and assessment of blood flow volume.
Line 231-234: please add reference.
Comments on the Quality of English LanguageThe English language is fine, only minor editing I think is needed.
Author Response
Reviewer 1
Comment (Line 25-26): Please indicate the examination with which the diagnosis and categorization of congenital portosystemic shunts was made. It is not clear that a CT scan was performed.
Response: Thank you for pointing this out. We have revised the sentence to clarify that the diagnosis of types of portosystemic shunts were confirmed using computed tomography angiography. The revised sentence now reads, “This study aimed to compare liver volumes in dogs with different types of PSS confirmed by computed tomography angiography, using CTHV.”
Comment (Line 32-33): Please indicate the age and body weight of dogs, an average value.
Response: Response: Thank you for your suggestion. The original sentence was based on our correlation analysis between liver volume and related parameters. As such, the data do not provide an exact age or body weight threshold for microhepatia, and we cannot specify an average body weight or age for dogs with microhepatia. Additionally, we identified an error in this sentence within the abstract. Throughout the rest of the manuscript, we used LV%diff as a measure to reflect the degree of microhepatia. Our analysis showed that age did not correlate with LV%diff, although it was significantly correlated with normalized liver volume (nLV). Therefore, age should not be considered a factor influencing or correlating with microhepatia. We have revised the sentence to focus solely on body weight as a factor affecting microhepatia: "Additionally, smaller dogs exhibited more severe microhepatia, with a significant positive correlation between LV%diff and body weight (p < 0.01)."
Comment (Line 40): Please refresh, non-invasive.
Response: Thank you for the feedback. The sentence has been revised for clarity, now reading, “Computed tomographic hepatic volumetry (CTHV) is a non-invasive volumetry method”
Comment (Line 105-106): Why wasn’t the same anesthesia or sedation protocol followed for all animals? Do you think that the use of different anesthetic drugs could affect the blood supply to the liver and therefore the total volume of the liver at the time of the examination?
Response: Thank you for raising this important point. As this was a retrospective study, the anesthesia protocol varied among the cases. We acknowledge that different anesthetic regimens could influence liver blood flow and volume during CT scans. To address this, we have now included the following in the discussion section: “One of the limitations of this study is the lack of a standardized anesthesia or sedation protocol for all cases. Because this is a retrospective study, the included dogs received different anesthesia and sedation regimens based on clinical need at the time of their respective CT studies. Variations in anesthesia protocols can affect physiological parameters such as hepatic blood flow and portal venous pressure, potentially affecting liver size measurements. The lack of a standardized protocol introduces an additional source of variability that may have affected the CTHV results. Future prospective studies using standardized anesthesia or sedation protocols would help minimize these confounding factors and provide more consistent data regarding hepatic volumetry in dogs with congenital portosystemic shunts.”
Comment (Line 220-226): Except from x-rays, it is necessary to mention the ultrasound examination for the liver size evaluation and the Doppler examination for the imaging of vessels and assessment of blood flow volume.
Response: We appreciate the suggestion. We have revised the sentence in the discussion to include ultrasound examinations as an additional method for evaluating liver size. However, we did not include Doppler examination here, as the focus of this particular sentence is on liver size assessment, rather than vascular structures or blood flow. The updated text reads: Assessing liver size through imaging modalities, such as radiography, ultrasonography, and CT, is essential for understanding various hepatic diseases in dogs [16-20].
Comment (Line 231-234): Please add reference.
Response: Thank you for your observation. A reference has been added, along with additional clarifications to prevent any confusion. The revised sentence now reads: “Although chronic inflammatory disease, fibrosis, cirrhosis, atrophy, and PSS are commonly considered to be causes of microhepatia in dogs[17-19], the poor accuracy of radiographic evaluation of liver size, due to variations in body conformation, body condition, and the influence of respiratory timing, has resulted in an incomplete understanding of the presence of microhepatia in dogs[21,22].”
Reviewer 2 Report
Comments and Suggestions for Authors
General comments
The objectives of this study are clear and the discussion is accurate. One point I would like to confirm is that the breed is not indicated in the results of this study, but is there a species difference in liver volume?
Please provide a breakdown of the breeds and a discussion on this point.
Specific comment
Line95-96:Although you write “Dogs with any hepatic abnormalities on the CT study were excluded from the present study.” what specific abnormalities are you referring to? Does it refer to the presence of a mass, etc.?
Author Response
General Comment: The objectives of this study are clear and the discussion is accurate. One point I would like to confirm is that the breed is not indicated in the results of this study, but is there a species difference in liver volume? Please provide a breakdown of the breeds and a discussion on this point.
Response: Thank you for your insightful comment. We have included the breed distribution in the results section as follows:
"The breeds included were 10 mixed-breed dogs, 9 Golden Retrievers, 8 Yorkshire Terriers, 6 Labrador Retrievers, 4 Pugs, 2 Maltese, 2 Miniature Schnauzers, and 1 each of American Cocker Spaniel, American Eskimo, Beagle, Bernese Mountain Dog, Border Collie, Cockapoo, Goldendoodle, Irish Setter, Irish Wolfhound, Labradoodle, Lhasa Apso, Miniature Pinscher, Norwegian Elkhound, Pembroke Welsh Corgi, Toy Fox Terrier, and Welsh Terrier."
We calculated the mean nLV and LV%diff for breeds with more than two dogs, and the values varied as follows:
- Mixed-breed dogs (n = 10): nLV 16.8 ± 5.3, LV%diff -60.3% ± 11.3
- Golden Retrievers (n = 9): nLV 14.6 ± 4.0, LV%diff -44.5% ± 3.4
- Yorkshire Terriers (n = 8): nLV 21.5 ± 7.8, LV%diff -69.0% ± 10.7
- Labrador Retrievers (n = 6): nLV 15.0 ± 4.1, LV%diff -36.3% ± 15.8
- Pugs (n = 4): nLV 15.5 ± 1.1, LV%diff -56.6% ± 1.6
- Maltese (n = 2): nLV 20.5 ± 10.7, LV%diff -57.9% ± 9.1
- Miniature Schnauzers (n = 2): nLV 15.6 ± 0.6, LV%diff -54.0% ± 7.4
We did not include these values in the manuscript due to the small sample sizes within each breed and the variability in shunt types. Discussing breed differences meaningfully would require grouping breeds by body conformation or size, which we have already addressed by discussing the effect of body weight on liver volume. A larger study focusing on specific breeds with consistent shunt types would be needed to assess true breed differences. We may pursue this as a future research direction.
Again, thank you for your suggestion.
Specific Comment (Line 95-96): Although you write “Dogs with any hepatic abnormalities on the CT study were excluded from the present study,” what specific abnormalities are you referring to? Does it refer to the presence of a mass, etc.?
Response: Thank you for your comment. We initially intended to exclude dogs with liver abnormalities unrelated to PSS that could affect liver volumetry, such as hepatic malignant neoplasia. However, none of the dogs in the study exhibited these abnormalities, so no exclusions were necessary. The sentence has been revised as follows: " Dogs with any hepatic disease unrelated to PSS that could affect liver volumetry, including hepatic malignant neoplasia, were considered for exclusion. However, none of the dogs exhibited such abnormalities, and thus, no exclusions were made."